# Sexual Exploitation as a Minor, Violence, and HIV/STI Risk among Women Trading Sex in St. Petersburg and Orenburg, Russia

**DOI:** 10.3390/ijerph16224343

**Published:** 2019-11-07

**Authors:** Lianne A. Urada, Maia Rusakova, Veronika Odinokova, Kiyomi Tsuyuki, Anita Raj, Jay G. Silverman

**Affiliations:** 1Department of Medicine, Center on Gender Equity and Health, University of California, 9500 Gilman Dr. MC0507, La Jolla, CA 92093-050, USA; ktsuyuki@ucsd.edu (K.T.); anitaraj@ucsd.edu (A.R.); jgsilverman@ucsd.edu (J.G.S.); 2School of Social Work, San Diego State University; 5500 Campanile Dr., Hepner Hall Room 119, San Diego, CA 92182-4119, USA; 3Department of Sociology, St Petersburg University, Universitetskaya Emb., 7–9, St. Petersburg 199034, Russia; rusakova.maia@yandex.ru; 4Sociological Institute, Federal Center of Theoretical and Applied Sociology, Russian Academy of Sciences, st. 7th Krasnoarmeyskaya, 25/14, St. Petersburg 190005, Russia; veronika.odinokova@gmail.com

**Keywords:** child sexual exploitation, human trafficking, sex trade, violence victimization, HIV, Russia

## Abstract

Child sexual exploitation (CSE) is a major risk factor for acquiring human immunodeficiency virus/sexually transmitted infections (HIV/STI), violence and other health concerns, yet few studies have examined these associations in Russia until now. This study examines the prevalence of CSE (those entering the sex trade as a minor) among women in the sex trade in Russia and how exposures and behaviors related to violence and HIV/STI structural risks differ from those who entered the sex trade as an adult. Women in the sex trade (*N* = 896) in St. Petersburg and Orenburg, Russia were recruited via time-location sampling and completed structured surveys. Adjusted logistic regression analyses assessed associations between CSE victimization and HIV risk-related exposures. Of the 654 participants who provided their age at first sexual exploitation, 11% reported CSE prior to age 18. Those who reported CSE were more likely to be organized by others and to be prohibited from leaving a room or house and from using condoms; three-quarters experienced rape when trading sex; a third were involved in pornography before age 18 and they had less education if they entered the sex trade as a minor. In adjusted analyses, those entering the sex trade as a minor were significantly more likely than those entering the sex trade as an adult to report drug use prior to age 18 (AOR = 5.75, 95% CI = 2.53–13.09) to have ≥5 clients/day (past 12 months; AOR = 3.55, 95% CI = 1.56–8.08), to report receiving police assistance (AOR: 3.10, 95% CI = 1.26–7.54), and to have fewer experiences of police extortion (AOR = 0.35, 95% CI = 0.10–1.24). They were four times more likely to participate in pornography before the age of 18 (AOR = 4.08, 95% CI = 1.32, 12.60) and three times more likely to have been sexually abused as child (AOR = 2.93, 95% CI = 1.27, 7.54). Overall, entry as a minor was related to greater risk for victimization and an inability to protect oneself from STI/HIV.

## 1. Introduction

An estimated 24.9 million are victims of human trafficking worldwide, and 4.8 million are sexually exploited [1]. Children, mostly girls, account for 30% of all trafficked persons [2]. The term sex-trafficked victim includes any person who is forced, coerced, or deceived into the sex trade [3]; those under 18 years old are referred to as Child Sexual Exploitation (CSE).

Although the literature on underage entry into the sex trade remains scarce, studies conducted in India, Canada, Mexico, Nepal, the Philippines, and Thailand have corroborated the health outcomes, safety risks, and consequences for minors who are sexually exploited. Victims brought into the sex trade as CSE had more vulnerabilities than persons who were sexually exploited as adults [4,5,6,7]. The following risk factors for minor entry compared to adult entry into the sex trade have been identified: greater biological vulnerability to HIV [8,9], increased likelihood of a genital track tear because of repeated physical trauma [10,11], greater vulnerability to violence [11,12,13], higher levels of sexual risk behaviors [10,11,14], lower levels of HIV knowledge [11], and a higher number of sexual contacts per day [15]. In Russia, HIV infection rates run higher than rates in other nations with a disproportionate prevalence of Russian women involved in the sex trade and injection drug use. Of these alarming statistics, little regarding HIV risk, drug usage, and health outcomes for CSE has been investigated.

While prostitution is legally prohibited in the Russia Federation, with ramifications amounting to 2000 rubles or approximately US$30 (Code of Russian Federation on administrative offenses, Article 6.11), legalities concerning minors remain undeveloped [16]. Article 240.1 specifies special punishment for those “obtaining sexual services of a minor (aged 16–18) for which is monetary or any other remuneration to a minor or third party or a promise of remuneration to a minor or third party” [16], yet language for those younger than 16 is nonexistent. The Russia Federation does not explicitly define child sexual exploitation.

The Russian Federation is ranked poorly for its efforts to eradicate human trafficking, having one of the worst global records [17]. The Russian Deputy Prime Minister of Internal Affairs estimated 150,000–500,000 persons are engaged in prostitution in Russia [18], with over 9000 registered crimes categorized as human trafficking-related in 2013 [19]; 35% of the victims of the crimes were minors [20]. A previous study of 15–17-year-old students in St. Petersburg found that 3.8% sold sex for money [21] and, similarly, in the Baltic Sea Regional Study on Adolescents’ Sexuality [22], 3.6% of all female adolescents reported they exchanged sexual intercourse for pay, highlighting the vulnerability of Russian adolescents to be sexually exploited [22]. Experts have found that vulnerable minors in Russia run a high risk for child pornography [23,24].

Involvement in the sex trade has been linked to HIV exposure and risks. In Russia, as high as 15–65% of women trading sex had tested positive for HIV infection [25]. Furthermore, drug use by injection has been reportedly more common among women involved in the sex trade relative to the general population; approximately 25–80% of women trading sex use drugs by injection [26,27] and a reported 48.1% of those with injection drug use have HIV infection [25]. However, less is known about the associations between CSE and HIV/STI risks in Russia.

In the last two decades, the number of people living in Russia with HIV have increased over 10-fold from 86,356 in 2000 [28] to 515,664 in 2010 [29], and more recently, to an estimated 1.2 million people in 2019 [30]. Of the latest population growth, 37% are women, 36% are on treatment, and 27% are virally suppressed [31]. Although injection drug use (IDU) has been the primary means of HIV exposure in Russia [31,32,33,34,35], heterosexual transmission of HIV in nearly half of all cases has been presently observed [31,36,37]. This shift means further research is required in the relationship between participants of the sex trade, injection drug users, and HIV risk, particularly in relation to CSE persons in Russia. The present study is the first to empirically examine the prevalence of CSE in the sex trade in Russia and how exposures and behaviors related to violence and HIV/STI structural risks differ for this population. The objective of this paper is to contribute to the broader understanding of the Russian epidemic context.

## 2. Materials and Methods

Data were collected in the Russian cities of St. Petersburg and Orenburg to characterize the conditions of the women trading sex in these two different urban areas. St. Petersburg is a center of culture and education and a major tourist city with a population size of over 5 million residents. In contrast, Orenburg is a city of approximately 600,000, bordering Kazakhstan and is an industrial area where gas and oil are mined and processed. In St. Petersburg, a majority of women trade sex on the streets independently and are dependent on heroin. Orenburg is characterized by more organized prostitution with fewer women who are dependent on drugs. 

Both cities have residents who are mostly of Russian ethnicity (85% and 74%, respectively). The 2010 All-Russian Census [38] is the last to collect data on more than 45 nationalities living in Saint Petersburg, including Ukrainians, Belarusians, Tatars, and Jews; altogether representing less than 2% of the population. In Orenburg, other nationalities are Tatars (7.6%), Kazakhs (6%), Ukrainians (2.5%), Bashkirs (2.3%), and others (less than 2%).

The ethical review board of the Sociological Institute of the Russian Academy of Sciences, Deviance and Social Control Department approved this study. To minimize risks and to ensure the confidentiality and safety of the participants and interviewers, we informed the sex-trade managers and police departments about the study and its conditions in advance. We also combined the invitation to participate in the study with the promotion of an outreach program, thus improving the participation and maximizing the potential benefits for study participants. Additionally, we pre-tested the questionnaire to ensure that the content and language matched the age, education, and experience of the target group. To minimize the risk of pressure from the managers, we trained the interviewers to pay special attention to the potential participants’ consent and capacity to make a rational decisions about the interview.

### 2.1. Sampling Procedures

This study involved cross-sectional quantitative data collected from women, ages 18 and up, in the sex trade (*N* = 896; June 2007 to March 2008) recruited from St. Petersburg and Orenburg. Sampling procedures were developed for each city to reflect the structural features of sex-trade activity, including location and type of sexual exploitation for each city. To connect location and type of sex-trade involvement, street-based women trading sex were sampled in particular due to their consistency in working at a particular location. For those who did not work under a manager, 20 people (3.0%) in St. Petersburg and two people (0.9%) in Orenburg indicated participating in the internet-based sex trade.

In St. Petersburg, recruitment sites were determined using a time-location sampling procedure created from key informants and observational inputs, i.e., a listing of locations street-based women traded sex by the time of their sex trade. Seventy-three time-location clusters (with different times/days across 15 locations) were identified for inclusion in the study with a probability of being selected proportional to the total number of women in the sex trade for each location. For non-street-based women, the NGO Stellit’s (non-governmental organization in Russia leading this study) included all venue and outdoor type sites based on relationships with managers at six brothels, two hotels and one railway station to facilitate recruitment. 

In Orenburg, the same time-location sampling process was used; however, all identified locations were included in the study. Twenty-five time-location clusters of street sex trade were identified; an average of five women presented at each site throughout the day. The Orenburg sample included street sites and three hotels. Interviewers visited each selected location in both cities 2–3 times; all available women trading sex were invited into this study. Women who were available solely via calls (e.g., call girls, escorts) were not included due to a lack of connection to a location. However, we recognize that women are recruited now more than ever over the internet, social media platforms, and via mobile phones in Russia, and that these are likely now the most prevalent means of procuring transactional sex in these cities.

### 2.2. Recruitment and Participation

Fieldwork was carried out by outreach program employees in St. Petersburg and Orenburg. The program included HIV risk education, provision of free condoms, assistance in obtaining state medical and social services, psychological assistance, and consultations with “trusted” doctors. At the same time, women were invited to become research participants. Trained research staff approached women directly at street sites and, with the approval of managers, approached them at venue sites. Participants were informed about the purpose of the study, which was a sociological examination of the sex trade and the social and health needs of women engaged in transactional sex in Russia. Six-hundred-and-sixty-five of 680 women trading sex in St. Petersburg who were invited to participate in this study consented (97.8% response rate), and 231 of 235 women trading sex in Orenburg consented (98.3% response rate), yielding a total of 896 participants. As noted earlier, the survey was combined with the recruitment of women into an HIV-prevention outreach program, which included HIV counseling and education as well as condom provision. This strategy likely explains the high response rate and participants’ motivation to participate in this study.

### 2.3. Study Procedure

After obtaining signed informed consent, all participants completed a structured face-to face survey interview developed for this study. Surveys were approximately 90 min long and assessed demographic profiles, substance use, sexual behaviors, sex-trade experiences, and other social and health issues relevant to women trading sex in Russia. Surveys were conducted in private locations. In street sex-trade settings, a van was often used to provide a private location for data collection. Participants received a small gift of cosmetics (equivalent to U.S. $18) for completing the surveys. 

### 2.4. Measures

Although several items included in this study were used in previous studies with women trading sex [39], prior to study implementation, the survey was pilot tested in St. Petersburg to ensure its clarity and utility with this vulnerable population. Demographic items assessed included the participant’s current age, number of children, education, and marital status. The variable “age when you had your first commercial sexual contact with a man in return for money, drugs or other sorts of compensation,” was dichotomized for separate groups—women under age 18 and women over 18. CSE was assessed for its association with the following: self-reported STIs (ever), involvement in sex trade organized by others or by oneself, and having five or more clients per day during the past 12 months, substance use before the age of 18 and in the past 30 days, and whether they had ever been victimized by childhood sexual abuse. 

The survey had multiple variables for HIV/STI risk. Participants were asked about their sex-trade experiences: “Have any of the following events listed in the table below happened to you during your involvement in sex [trade] business?” with response choices: rape; prohibition to go out of some house or room into the street; giving you a sexually-transmitted infection; prohibition to use personal contraceptives and means of protection from STIs; and refusal to provide you with necessary medical help. They were also asked whether their sex-trade involvement was organized by others and whether they participated in pornography before the age of 18 (being photographed naked or alone for payment). We chose CSE as a dependent variable to determine how exposures and behaviors related to HIV/STI risk differ based on the experience of CSE.

Interaction with police was measured by the question “Did your contact with police during the last 12 months involve any of the following events?” Responses were collapsed into the following four categories: (1) Assistance: protection from clients, protection from violence on the part of organizers of your business, protection from incidental crime or sexual abuse, help in freeing you from organizers of your business, help in getting or reissuing necessary documents, and moral support; (2) Extortion of money you earn; (3) Coercion: to give out information about clients, or to render sexual services, free of charge; (4) Work coercion: coercion to do unskilled physical labor for the police, free of charge. 

### 2.5. Data Analyses

Descriptive analysis of all variables was conducted for the total sample and stratified based on minor age at first sex exploitation and adult age at first sex-trade act. Chi-square analyses were conducted to assess associations between all variables and age at first sexual exploitation. A stepwise approach was used in multivariable regression models to determine associations between HIV-related exposures and age at first sexual exploitation. All variables significantly associated with age at first sexual exploitation in chi-square analyses were considered for inclusion. As a first step, collinearity between demographic variables was assessed via Spearman correlation analyses; correlations of 7 or greater were viewed as collinear so that only age, education, and having minor-aged children were included as control variables in the final multivariate model. Adjusted logistic regression analyses assessed associations between minor age at first sexual exploitation and HIV-related exposures. We controlled for their duration in the sex trade with the question “How long have you been selling sex altogether starting from your first commercial sex contact?” to control for the fact that longer periods of sexual exploitation are experienced by those who enter the sex trade as a child. 

## 3. Results

Table 1 describes the percent distribution of demographics, substance use, and sex-trade characteristics of the women selling sex in St. Petersburg and Orenburg, Russia. The majority of women were ages 20–24 (31%) and 25–29 years old (43%), most had a secondary school education, only 10% were married, and 32% had minor-aged children. Although only 19% used drugs before age 18, 68% reported drug use in the past 30 days. Forty-one percent reported ever having a sexually transmitted infection (STI). In terms of experiences in the sex trade, about one-third reported that their sex-trade involvement was organized by others (34%) and had a daily average of five or more clients over the past 12 months (34%). The majority reported experiencing rape (67%). Also, 36% were prohibited from leaving their room/house, while 37% were prohibited from using condoms or contraception. In terms of police interaction, three-quarters of the women (75%) reported some type of police contact in the past 12 months, with 63% reporting police extortion, 19% reporting police coercion, 43% reporting police work coercion, and only 21% reporting having received assistance from police.

Of the 654 women trading sex who provided their age at first sexual exploitation, 11% reported being under the age of 18 at the time they entered the sex trade. Forty-three percent of those who entered the sex trade as a minor said their first encounter with sexual exploitation was “rather forced” instead of voluntary (57%) compared to those who entered the sex trade as an adult who felt it was rather forced (53%) vs. rather voluntary (47%). Compared to those entering as adults, many more had experienced child sexual abuse and experiences with pornography before the age of 18. Seven percent of those who entered the sex trade under the age of 18 were migrants from outside of Russia. For those who did not remember their age at first sexual exploitation (27% of 896), their characteristics did not differ significantly from the 654 individuals included in this analysis (e.g., on age, education, being managed by others, police contact, length of time trading sex, drug use past 30 days and ever, child sexual abuse victimization, pornography, immigrants from outside of Russia, restricted mobility, prohibited from using contraceptives, rape during sex trade). Women trading sex who entered the sex trade as minors were significantly younger at the time of survey, less educated, and less likely to be married and have minor-aged children. Women who entered the sex trade as minors were significantly more likely to report drug use prior to age 18 (not including alcohol) (47%; OR: 5.00, 95%CI: 2.97–8.43), but were less likely to use drugs in the past 12 months (50%; OR: 0.43, 95%CI: 0.26–0.70). Those who entered the sex trade as minors were significantly more likely to engage in sex trade organized by others, have an average of five or more daily clients, have their movement limited, and be prohibited from using contraceptive protection against STI/HIV. Those who entered the sex trade as minors also had significantly less contact with police, including fewer experiences with police extortion, coercion, and work coercion than women who entered the sex trade as adults. Twenty-nine percent of those entering the sex trade under the age of 18 participated in pornography before the age of 18. 

In multivariable regression (Table 2), those entering the sex trade as a minor were significantly more likely than those entering the sex trade as adults to report drug use prior to age 18 (AOR = 5.75, 95%CI = 2.53–13.09). They were four times more likely to have ≥5 clients/day (past 12 months; AOR= 3.55, 95%CI = 1.56–8.08). Those entering the sex trade as a minor were also more likely than those entering the sex trade as adults to report receiving police assistance (AOR: 3.10, 95% CI = 1.26–7.54) and having fewer experiences of police extortion (AOR = 0.35, 95%CI = 0.10–1.24). They were four times more likely to participate in pornography before the age of 18 (AOR = 4.08, 95%CI = 1.32, 12.60) and three times more likely to have been sexually abused as child (AOR = 2.93, 95% CI = 1.27, 7.54).

## 4. Discussion

One in nine women in this Russian study were victims of child sexual exploitation. Entry into the sex trade as a minor was related to less ability to protect oneself from STI/HIV. Those who traded sex as a minor were six times more likely than those entering sex trade as adults to report having used drugs before the age of 18. They were less likely to have technical or college education and were four times more likely to have a greater number of clients per day. Findings also revealed structural barriers to their protection from force/violence and HIV. Compared to those who entered the sex trade as adults, they were prohibited from leaving the house and using contraceptives to protect themselves. However, those entering the sex trade as a minor reported receiving more police assistance and fewer experiences of police extortion in adjusted analyses. The results of this study provide insights into the risks for those who entered the sex trade as adolescents in two Russian cities.

The women in this study in both St. Petersburg and Orenburg experienced a high prevalence of sexual violence and other types of force. These findings are consistent with other studies of women in the sex trade in Russia [40,41]. Overall, nearly three-quarters of the women who entered the sex trade as minors in this study reported experiencing rape (from anyone in the previous 12 months). More than two-thirds of those who entered as minors were prohibited from leaving their room/house or from using condoms or contraception. These results coincide with other studies about trafficked individuals and minors trapped in brothels [9,42]. Studies in other countries have found that youth who trade sex are especially vulnerable to HIV and related risks, including violence, less access to assistance, and higher chances of others controlling their sex-trade involvement [7,43,44]. In a Baltic Sea Regional Study on Adolescents’ Sexuality [22], 20–36% of Russian adolescent girls aged 16–18 reported having experienced any kind of non-penetrative sexual offenses, and 6–13% reported having experienced sexual abuse in the form of sexual intercourse. The average age of first sexual assault was about 14 years old. It was common that the offender was more than 5 years older than the victim. In this study, being a victim of childhood sexual abuse was associated with a three-fold likelihood of CSE. Our findings have important implications for the trauma these women may accumulate from their past abuse.

In this paper, those entering the sex trade as minors were also more likely to have five or more clients/day during the past year, which might be explained by their lower educational levels compared to those who entered as adults. Women who experienced CSE, compared to those who did not, may have taken a different pathway to prostitution after dropping out of the educational system early. Trading sex may be their only means of survival and income, resulting in their need for a higher number of clients per day in the sex trade. Trafficked youth are usually deprived of opportunities to receive an education and to develop emotional skills to facilitate healthy behaviors, responsible attitudes within their personal lives, and an aversion to risky behaviors [45,46]. Although having more clients might result in a higher risk of contracting STIs, those who entered the sex trade as minors did not have significantly more STIs when adjusting for their duration in the sex trade. However, research shows that Russian children who are sexually exploited are particularly vulnerable to STIs; by age 14, sexually exploited children in the Baltic study had a significantly higher incidence of syphilis and other STIs than adults [47]. Prohibition from using condoms or contraceptives might also better explain STI risk. 

In the current study, nearly a third of those who experienced CSE had participated in pornography before age 18, and many more who said they did not trade sex before age 18 said they did participate in pornography before age 18. The abovementioned Baltic study revealed a lack of protective attitudes towards sexual exploitation among Russian adolescents, which might explain their vulnerability. Many Russian adolescents (both girls and boys) supported attitudes in favor of sex between an adult and a child. The majority was inclined to think victims who let others sexually exploit them were responsible for the exploitation [22]. Female adolescents aged 16–18 reported that they could be involved in intercourse for a reward (5–10% of different ages) and would have oral sex for a reward (from 2–8%) or anal sex for a reward (from 1–4%). More female adolescents said they could accept someone having intercourse for a reward compared with those who were not willing to accept it (53–63%) [22]. These attitudes may set them up for vulnerability to entering pornography at an early age and should be further explored. Perhaps it is not surprising that over half in the present study perceived their first CSE as rather voluntary instead of forced.

Further increasing their vulnerability, substance use was highly associated in this study with entering the sex trade as a minor; they were eight times more likely than those entering the sex trade as adults to report drug use before the age of 18. To facilitate child participation in acts of a sexual nature, children and adolescents are often encouraged to drink alcohol and take drugs or other intoxicating and mind-altering substances in Russia [48]. Previous analyses in Russia found an association between CSE and drug use, early age of first sex, and factors associated with their family of origin (e.g., conflicting relationships between parents, family structure other than a complete family with both natural parents, and a marital betrayal by the father) [48]. More research is needed in Russia on whether substance use dependency among young people is a pathway to entering the sex trade and how it might be prevented.

In terms of police interaction, three-quarters of the women reported having some type of police contact in the past 12 months, with the majority experiencing police extortion or coercion, but less than a quarter receiving police assistance. Those entering the sex trade as a minor, however, were more likely than those entering the sex trade as adults to report receiving police assistance and experiencing less police extortion. For this study, it is likely that when the women were under 18, they had more assistance and less extortion from the police because they were less involved in drug use. In previous research, we demonstrated that drug-using women in the sex trade are most likely to be coerced by police, probably due to the double stigma [49]. Police may treat those under 18 as victims and less as ‘professional’ prostitutes.

There are several limitations to this study worth noting. Due to the cross-sectional nature of the baseline data, causation among variables cannot be inferred. The study is also limited by the potential bias of collecting self-reported data if the participants’ recall may be inaccurate or their responses influenced by a social desirability bias. Also, age was not collected as a continuous variable, thereby restricting our knowledge of the participants’ ages to categories only. The age of first sexual exploitation was also limited to the categories above age 14, and the age of being a victim of childhood sexual abuse (genital sexual contact) was also not specified so it is possible these two categories overlapped. The time of generalizability may also be limited by the time and location of the data collected at the study sites in the two Russian cities. Like many other global contexts, sex-trade activity has become less visible in urban public spaces, with exchanges increasingly made online and via mobile phones in these two Russian cities. Also, the routes of HIV transmission (injecting drug use) and specific Russian laws discussed in this paper are not generalizable to women in the sex trade in every other national settings. A small number of participants migrated from countries outside of Europe, but the countries were not specified. Furthermore, men were not included, though research on sex trafficking of males and men who have sex with men is needed, especially in the context of HIV transmission risk rising among these populations globally and in Russia.

Nevertheless, the findings may help us understand the contexts in which women who entered the sex trade at an early age endured and how to tailor interventions for them based on the trauma and lack of autonomy they experienced. Also, in many other global contexts, women and minors continue to exchange sex in brothels, hotels, bars/night clubs, massage parlors, and on the street where they are subject to force/coercion, violence, deception, and risk [50,51,52,53,54]. Little has changed over the past decade in terms of the institutional forms of social control of prostitution in Russia and the social assistance offered to these women. 

## 5. Conclusions

Although little data exists on the number of children involved in the sex trade in Russia, studies, including this one, have documented a significant number of children who were sexually exploited in Russia [22,38,55,56]. Findings from this study confirm the need for policy and practice in Russia to target adults and minors separately. Globally, adolescents are more vulnerable due to their lack of inhibition, risk-taking behaviors, and tendency to plan for their futures less [57]. With sex education prohibited in Russia, other ways to prevent sexual exploitation and health risks among children are necessary because of the vulnerabilities youth face from adults (known or unknown to the child) and from peers that may lead them to a pathway of child abuse, substance use, and sex-trade involvement (e.g., in organized crime syndicates in Orenburg, or working in venues or for drugs or partners in St. Petersburg).

In the United States, measures such as the Safe Harbor Law (prohibiting sentencing of trafficked minors) [58] and the creation of specialty courts for human trafficking-involved minors (with a joint collaboration between child welfare and law enforcement) have been adopted in some states. However, in Russia, mechanisms to prevent child sexual abuse (including exploitation in the sex trade), identification of it, and assistance to victims have not yet been created at the state level [59]. Therefore, the general awareness of adults (e.g., parents and social workers) about child sexual abuse in Russia is low, which contributes to a low detection of such cases. Though Russia established a child protection system in 1999, no state body in Russia coordinates the activities of various departments involved in protecting children from sexual exploitation and sexual abuse [60]. Thus, children at risk of sexual exploitation (or those who are already involved) have a high chance of being on the child protection system’s radar, but a low chance of being identified as victims of sexual exploitation and getting the help that matches their needs [61]. Screening tools could be adopted such as the Commercial Sexual Exploitation-Identification Tool (CSE-IT) [62] or others [63,64]. The CSE-IT is a universal screening tool that has assessed U.S. youth aged 12 and above in the child welfare system by asking caseworkers to identify eight areas with indicators of the early risk signs for child sex trafficking or exploitation: housing and caregiving; prior abuse or trauma; physical health and appearance; environment and exposure; relationships and personal belongings; signs of current trauma; coercion; and exploitation.

In 2013, Russia ratified the Council of Europe Convention on the Protection of Children against Sexual Exploitation and Sexual Abuse (CETS No. 201) of 25 October 2007 [65]. A complex process is currently underway to implement the provisions of the Convention in the criminal law of signatory countries. After ratification of the Convention, some laws aimed at implementing its provisions were adopted, but they were not enough to fulfill its requirements regarding the prevention of sexual abuse against children [65]. A number of the Convention’s provisions require an adequate response of the Russian legislature to introduce amendments and additions to the legislation of the Russian Federation to effectively prevent the sexual exploitation of minors.

Our study shows that state systems need to effectively identify and assist minors who have been involved in sex trafficking, including psychosocial rehabilitation and measures to halt recidivism. This process should include several stages: identification, emergency assistance, stabilization, and re-integration. Stakeholders (psychologists, social workers, juvenile inspectors, etc.) need training to detect and prevent sex trafficking of high-risk groups, especially children who use drugs and other vulnerable groups of youth. Child pornography should also be viewed as a form of human trafficking and sexual exploitation.

Finally, data for this study were collected before the implementation of a 2012 Russian Federation federal law that prohibits the portrayal and description of sexual acts of children under 16 years old to “protect children from information harmful to their health and development” [66]. Also, data were collected on those aged 18 and above. According to the law, any questions about the sexual experience of children under the age of 18 (the age of consent under Russian law) can be interpreted as an incitement to sexual activity. The policy poses a barrier to conducting future population-based studies on the experiences of sexually-exploited children in Russia because inquiries into surveys about sexual encounters for those under age 18 may be interpreted as inciting sexual activity. Therefore, the ability to conduct studies on CSE in Russia is now rare. 

## Figures and Tables

**Table 1 ijerph-16-04343-t001:** Socio-demographic and behavioral characteristics of women in the sex trade in St. Petersburg and Orenburg, Russia (*N* = 654).

Characteristics	Percentage Distribution ^¥^					
	Total Sample		Entered ≥18 Years	Entered <18 Years	*p*-Value	Odds Ratio (95% CI)
**(Socio-demographic & Behavioral)**	100%	*N* = 654	89% (*N* = 584)	11% (*N* = 70)		
Demographics Age						
18–19	5	33	3	26	***	ref.
20–24	31	198	29	41		0.14 (0.65, 0.32)
25–29	43	275	44	29		0.07 (0.29, 0.15)
30+	22	139	24	4		0.02 (0.01, 0.70)
Education						
Secondary	53	345	52	67	*	ref.
Vocational/Professional	21	135	21	17		0.62 (0.32, 1.21)
Technical/College/University	26	165	27	16		0.45 (0.23, 0.90)
Married	10	67	11	4		0.36 (0.11, 1.17)
Has minor-aged children	32	208	35	13	***	0.28 (0.14, 0.57)
The main place brought up						1.19 (0.90, 1.59)
Parents	91	593	92	89		
Relative’s family	41	27	42	4		
Foster family	1	9	2	0		
State-sponsored institution	21	14	21	20		
Family & state-sponsored place	2	11	1	1		
Sexually abused as a child (ever)	16	146	13	34	***	3.51 (2.04, 6.14)
Substance use						
Drug use before age 18 (excluding alcohol)	19	120	15	47	***	5.00 (2.97, 8.43)
Drug use (past 30 days)	68	438	70	50	***	0.43 (0.26, 0.70)
STI Ever (self-reported)	41	262	40	49		1.44 (0.87, 2.36)
Experiences in sex trade (ever)						
Sex trade organized by others	34	218	31	60	***	3.40 (2.04, 5.66)
Avg. ≥5 clients/day (past 12 months)	34	208	31	46	**	1.91 (1.15, 3.16)
Raped	67	429	66	74		1.52 (0.86, 2.66)
Prohibited from leaving the room/house	36	230	34	47	*	1.71 (1.04, 2.82)
Prohibited from using protection	37	237	35	49	*	1.73 (1.05, 1.85)
Participated in pornography before 18	9	58	3	29	***	0.20 (0.11, 0.37)
Was refused medical help	12	78	12	10		0.80 (0.35, 1.82)
Interactions with police						
Contact with police (past 12 mos)	75	486	77	64	*	0.55 (0.32, 0.93)
Received assistance from police	21	135	21	24		1.24 (0.69, 2.22)
Police extortion	63	405	66	40	***	0.35 (0.21, 0.58)
Police coercion	19	124	21	6	**	0.23 (0.08, 0.64)
Police work coercion	43	280	46	21		0.32 (0.18, 0.58)

*** *p*-value ≤ 0.001, ** *p*-value ≤ 0.01, * *p*-value ≤ 0.05; ^¥^ Percentages rounded & columns may sum to >100%.

**Table 2 ijerph-16-04343-t002:** Adjusted odds of being a minor at first sexual exploitation of Russian women in the sex trade in St. Petersburg and Orenburg, Russia (*N* = 645).

Characteristics	Adjusted OR ^†^	95% CI
Drug use before age 18	5.75	(2.53, 13.09) ***
Drug use (past 30 days)	0.44	(0.17, 1.15)
STI Ever (self-reported)	1.87	(0.85, 3.89)
Sexually abused as a child (ever)	2.93	(1.27, 7.54) *
**Experiences in sex trade or CSEC**		
Sex trade organized by others	1.48	(0.57, 3.64)
Avg. ≥5 clients/day (past 12 mos)	3.55	(1.56, 8.08) **
Raped	1.207	(0.45, 2.52)
Prohibited from leaving room/house	0.98	(0.44, 2.18)
Prohibited from using protection (contraception)	0.71	(0.30, 1.65)
Participated in pornography before 18	4.08	(1.32, 12.60) *
**Interactions with police**		
Received assistance from police	3.10	(1.26, 7.54) *
Police extortion	0.20	(0.07, 0.62) **
Police coercion	0.35	(0.10,1.24)
Police work coercion	0.81	(0.28, 2.34)

^†^ Adjusted for age, education, having minor-aged children, and duration in the sex trade. *** *p*-value ≤ 0.001, ** *p*-value≤0.01, * *p*-value ≤ 0.05.

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
