# Peer review of "Sexual Exploitation as a Minor, Violence, and HIV/STI Risk among Women Trading Sex in St. Petersburg and Orenburg, Russia"

_ijerph, 2019, doi:10.3390/ijerph16224343_

Round 1

Reviewer 1 Report

Background:

The background section needs more organization and context.  Topic sentences need to be more clear and comprehensive. You should talk about CSE trends and consequences worldwide and then talk specifically about Russia. Then describe the gaps in research and the purpose of this study.  (I see you mention the purpose of the study more than once in the background.)

Also, what do we know about childhood sexual exploitation and its association with sex work and HIV? This has not been established in your background.

The description of CSE in Russia does not give regional or international points of reference so the reader cannot tell if Russia is below or above those averages for example.

You need to describe exactly how the policy makes research more difficult...is it because the government wont fund these studies or you cannot get people to participate? Please be specific. And how was this study able to get around those barrier?

The policy makes it difficult for 56 researchers to conduct population-based studies on the experiences of sexually exploited children in 57 Russia. 

I think it is unnecessary to define the term exploitation in the back ground.  Consider omitting this sentences: The term “exploitation” refers to the unfair use of someone for one’s own advantage or benefit, including both monetary and non-monetary exchanges [3].

What is the legal status of prostitution in Russia?

Methods:

Was the Orenburg sampling method also time-location sampling?  If so, please indicate how many time-location clusters were identified.

You mention internet based sex trade.  Do you know if those recruited for this study also participate in any internet based sex trade?

What was the approximate value of the small gift of cosmetics?

There was no section in the methods on ethical considerations. This is an extremely vulnerable population.  Readers should be made aware of the precautions you took to protect their safety and health.

Results

The tables needs to be formatting so that it is easier to read (bolded topic titles, etc.)

The description of the survey questions do not see to match what is in the tables. For example you mention many detailed questions related to police contact but then summarize in just a few item in the table. How does the list relate to the table?

The titles of the tables need to be more succinct and specific.

Discussion:

The findings of the study should be summarized succinctly in the first line of this section.

The limitation section sounds more like a defense of the paper. This should be a place where you offer transparently all the limitations.  Do you suspect other issues such as self-reporting bias, social desirability bias, or things relating to the specific Russian context or from those 2 cities that may be different than in other settings?

This paper needs to be proof read for grammar and sentence structure (run on sentences, missing articles, etc.)

Author Response

Reviewer 1:

Per the Reviewer’s very helpful recommendations for the Introduction/Background:

We have re-organized the content of the Introduction and made topic sentences of the paragraphs more clear and comprehensive. We added CSE trends and consequences worldwide and then talked specifically about Russia, described the gaps in research and ended with the purpose of this study.  We more thoroughly described what we know about childhood sexual exploitation and its association with sex work and HIV. We added to the description of CSE in Russia more regional or international points of reference so the reader can tell if Russia is below or above those averages. We moved to the conclusions the sentence about the Russian law that prohibits asking questions about sexual topics for those under age 18. According to this law, any questions about the sexual experience of children under the age of 18 (the age of consent under Russian law) can be interpreted as an incitement to sexual activity. We explained that data collection for this study occurred before the law went into effect. Also, all participants were ages 18 and over. We removed the “exploitation” definition in the introduction. We added a paragraph describing the illegal status of prostitution in Russia and its contexts and consequences for those involved in prostitution.

For the Methods:

We explained that the Orenburg sampling method was also time-location sampling and that 25 locations of street sex work were identified where an average of 5 women presented at each site throughout the day.  We explained that in Saint Petersburg it is 20 people (3.0%), and in Orenburg 2 (0.9%) were involved in internet-based sex trade. We added that the value of the small gift of cosmetics was equivalent to $18. We added a section on ethical considerations that included the precautions taken to protect the participants’ safety and health. (Lines 231-240).

For the Results:

We reformatted the Tables to make them easier to read (bolded topic titles, etc.). We reformatted the Measures section in the Methods section to explain how the variables for police interaction were combined to create the variables displayed in the tables. (Lines 387-394) We made the titles of the tables more succinct and specific as recommended.

For the Discussion:

We changed the first paragraph to reflect the main findings of the study with a first sentence summarizing succinctly this section. We added more to the limitations section to offer transparently all of the limitations, including self-reporting bias and social desirability bias. Our Russian authors said nothing else specific in the Russian context limited the generalizability of the results of this study. However, we did add that the routes of HIV transmission (injecting drug use) and Russian laws are not generalizable to women in the sex trade in some other national settings. Also, males or those trafficked to Russia from other countries were not part of the sample. In response to the reviewer’s suggestion, we checked this paper and revised it for grammar and sentence structure (run on sentences, missing articles, etc.).

Reviewer 2 Report

Dear Editors, Dear Authors,

congratulations on presenting a very interesting topic, how difficult and important from the point of view of public health. The subject matter is rarely discussed in the literature, and as needed to preserve the health of people put in such difficult life situations. The scientific article presents great theoretical values, but also important from a practical point of view. The scientific article was prepared with great care and meets the requirements set by the Scientific Journal.
            Particularly noteworthy are: elaboration of research results, discussion based on the latest literature and final conclusions significant for the subject matter discussed.

I propose to accept this scientific article as it stands.

Author Response

Reviewer 2:

Thank you very much for the very positive comments from Reviewer 2!

Reviewer 3 Report

This article provides essential knowledge to the general public and professionals working with youth on the issue of child sexual exploitation. It benefits from a strong research design, professionalism in handling sensitive information in a social and legislative context where discussions of the issue at study seem "prohibited"; it therefore constitutes a good contribution for public awareness on CSE and risk for HIV/STD. On page 2, the authors should clarify how they explain the conflict of information in paragraph 2 and 5: there were youth participants (see table 1 on page 5), how did the authors obtain ethical approval while apparently "violating the law" that prohibits discussion of sexuality of minors? Clarifying this might be helpful for other researchers who share similar research interests in Russia. Paragraph 1, on page 7, the authors should revisit the apparent correlation being drown between having an STI and number of clients per day as "restriction of condom use" seems more relevant based on the content of this article than the number of clients per day.

Author Response

Reviewer 3:

Reviewer Comment: The authors should clarify how they explain the conflict of information in paragraph 2 and 5: Did having youth participants “violate the law”?

Author Response: We clarified in the Methods and Tables that the participants were ages 18 and up. The survey only gave categorical age choices but the inclusion criteria was 18+ years old. We mentioned in the Discussion how the law in Russia didn’t go into effect until after data collection for this study ended.

Reviewer Comment: Paragraph 1, on page 7, the authors should revisit the apparent correlation being drown between having an STI and number of clients per day as "restriction of condom use" seems more relevant based on the content of this article than the number of clients per day.

Author Response: We recognize this as an important observation, so we added on Line 181-182 “prohibition from using condoms or contraceptives might better explain STI risk” and focused the paragraph more on the potential correlation of having lower education as an explanation of more client per day.

Reviewer 4 Report

Comments for the authors

General comment

This article is exploring the association between Child sexual exploitation and exposure to HIV and other STIs among women in sex trade in Russia. It aims to give prevalence figures for child sexual exploitation among women in sex trade and to show how women who experienced abuse as a child are more at risk for HIV and other STIs than the others. The article is very well written and it presents very valuable data on a marginalised, vulnerable population, i.e. women in sex trade in Russia, in a context where the HIV epidemic in Russia is exploding.

A few points are really worth improving nonetheless, in particular the analytical strategy chosen by the authors needs clarification, as well as the interpretation of some results.

Detailed comments

Abstract.

I was a bit disturbed by the term “female” throughout the text (but that could be because I am not an English native speaker), wouldn’t it be better to talk about women? The objectives are not clearly stated in the Abstract, although they are in the main text. For a reader who is not familiar with the Russian context, it is a confusing to read the reference to the police line 24, “Those who reported CSE were more likely than those who did not to be organized by others (e.g., police)” I would take out the parenthesis, it is explained in the text afterwards.

Introduction

The Introduction is clear and very well written. Maybe a few elements should be added : Another reference (in English) about how the epidemic in Russia may also be driven now by heterosexual transmission. I think this is an important point and the only reference the reader has is in Russian (ref 29). As the authors point out, the associations between CSE and HIV/STI exposures are well documented. Maybe it would be interesting to say why it is interesting to study the Russian context: in this way, the article is not “only” about re-examining know associations but contributing to a broader understanding of the Russian epidemic context.

Material and Methods

The authors say that the survey took place in St Petersburg and Orenburg to draw a more “representative picture” of women involved in sex trade in Russia. I think it would be more appropriate to talk about picturing the diversity of these women’ situations. On this point, it would be interesting to know a bit more about the choice of these two locations: does sex trade have different characteristics in the two cities?

However, thanks to the time location sampling, the authors could say that the sample is representative of women in sex trade in these two cities

The response rate is surprisingly high. Of course it’s a good thing but it is so high that it raises questions. Maybe the authors could bring some justification for it (long implantation of the outreach program maybe?) One of my main points is about the analytical strategy the authors used: as the objectives of the paper are stated, I was expecting a modelisation of HIV/STI exposure, with the variable CSE yes/no as the variable of interest (to assess whether the fact of having experienced CSE increases the probability to be exposed). I am thinking maybe because there were a lot of different types of exposures, the authors chose to modelise the probability to have been abused as a child and have the exposures as explaining variables.

However, I think this makes the interpretation of the results a bit more difficult (as I explain in my comments of the Results section). Could the authors explain a bit more why they chose this strategy and how they think it answers to their objectives?

Results

The only demographic characteristics that distinguish the women who experienced CSE and the ones who did not are Age and Education. Education is difficult to interpret because the causality could be in the other way (the women dropped their studies because of CSE). Does the difference in Age at time of survey relate to the fact that the CSE has been increasing in the recent years? Are there in the survey other characteristics that could help the reader to better understand how women who experienced CSE differ from the others? Are there more often foreign? After adjustment, the variables that remain significantly associated with the fact to start sex trade before 18 years old are: Drug use and pornography before 18years old (but since these two variables also refer to what happened before 18 years old it’s logical) A higher risk of ever having acquired STIs. On this, could the authors clarify whether they adjusted the multivariate regression on duration of trade sex? It is said in the Methods section but does not appear in any of the result tables. It could help to know what the median duration in sex trade is, whether it is very different between the two categories of women, and whether the association with experience of STI holds after taking the duration of sex trade into account The result which seems the most robust is the higher probability to have experienced CSE when one is having 5 or more clients a day in average. This is a well-known risk for STI and maybe the authors could insist more on this. The interactions with the police.

Discussion

I think the Discussion really needs to be edited to avoid any misinterpretation of the results: In particular, line 225-226 it is said: “These individuals more commonly reported not operating independently, and those organizing their sex trade were police, similar to other studies intersecting with police in Russia”. However, if I am not mistaken, the aOR is not statistically significant for this variable (1.14 [0.48-2.70]), plus the readers do not know how the authors know that these “others” would necessarily be the police. Another example line 238-239: “In this paper, those entering sex trade as minors were twice as likely to have an STI….” Those entering sex trade as minors were actually twice as likely to ever have an STI, which is not the same at all (please see above my comment on taking the duration in sex trade into account). Is there any explanation why the women who experienced CSE would have more clients a day than the others? Does that relate to different pathways into sex trade? Could the authors clarify whether “drug use before 18” comprises alcohol? There is a mistake on reference 44 line 264: the ref 44 does not correspond to a study in Russia It remains unclear for the reader how to interpret the results on the interactions with the police. In particular I didn’t understand why on the one hand the authors say that police was organizing the sex trade of women who entered sex trade as minors and on the other hand, that the same women declare less extortion from the police. Could the authors clarify that? As long as the interpretation of results remain uncertain regarding STI risk and interactions with the police, it is difficult to see how the conclusions are related to the results of the paper. As it stands, it constitutes an interesting discussion on policy implications on the topic of minors’ sex trade, however for now it is not clearly related to the data and results at hand.

Author Response

Reviewer 4:

Thank you for the thoughtful recommendations to clarify the analytical strategy and interpretation of some results. We clarify them below:

For the Abstract:

As suggested, we replaced the term “female” with “women” throughout the paper. We appreciate this feedback and believes it sounds better. We changed the objectives to be more clearly stated in the Abstract as they are in the main text. In reference to the police, Line 24, “Those who reported CSE were more likely than those who did not to be organized by others (e.g., police)” we took out the parenthesis as suggested as it is explained in the text afterwards.

For the Introduction:

We added another reference (in English) about how the epidemic in Russia may also be driven now by heterosexual transmission. We try to emphasize more the associations between CSE and HIV/STI exposures have not been well documented in the Russia context, and this paper therefore contributes to a broader understanding of the Russian epidemic context. Lines 91-107.

For the Material and Methods:

Reviewer Comment: The authors say that the survey took place in St Petersburg and Orenburg to draw a more “representative picture” of women involved in sex trade in Russia…Does the sex trade have different characteristics in the two cities?  

Author Response:  In St. Petersburg, a majority of women trade sex on the streets independently and are dependent on heroin. Orenburg is characterized by more organized prostitution with fewer women who are dependent on drugs. (Line 225)

Reviewer Comment: However, thanks to the time location sampling, the authors could say that the sample is representative of women in sex trade in these two cities.

Author Response: Thank you for this nuanced recommendation. We worded the following to reflect that representativeness is within the cities: “Data were collected in the Russian cities of St. Petersburg and Orenburg to characterize the conditions of the women trading sex in these two different urban areas.” (Line 109)

Reviewer Comment: The high response rate raises questions. Maybe the authors could bring some justification for it (long implantation of the outreach program?)

Author Response: We added that the survey was combined with the recruitment of women into an HIV prevention outreach program, which included HIV counseling and education as well as condom provision. This strategy likely explains the high response rate and participants’ motivation to participate in this study. (Line 359)

Reviewer Comment: One of my main points is about the analytical strategy the authors used: as the objectives of the paper are stated, I was expecting a modelisation of HIV/STI exposure, with the variable CSE yes/no as the variable of interest (to assess whether the fact of having experienced CSE increases the probability to be exposed). I am thinking maybe because there were a lot of different types of exposures, the authors chose to modelise the probability to have been abused as a child and have the exposures as explaining variables.

Author Response: Yes, we didn’t have a single proxy variable for HIV/STI risk, but rather multiple ones. We chose CSE as a dependent variable to address how exposures and behaviors related to HIV/STI risk differ based on the experience of CSE.

For the Results:

Reviewer Comment: The only demographic characteristics that distinguish the women who experienced CSE and the ones who did not are Age and Education. Education is difficult to interpret because the causality could be in the other way (the women dropped their studies because of CSE).

Author response: Actually, having minor-aged children was also adjusted for, as well as length of time selling sex (We re-ran the models adjusting for length of time selling sex). Being married or not was also included in Table 1.

Reviewer Comment: Does the difference in Age at time of survey relate to the fact that the CSE has been increasing in the recent years?

Author response: We are not sure how to answer this using this particular previously collected cross-sectional data. 

Reviewer Comment: Are there in the survey other characteristics that could help the reader to better understand how women who experienced CSE differ from the others? Are there more often foreign?

Author response: As reported, only a small percent (7%) were migrants from outside of Russia that took part in this data collection. We included the demographic characteristics that were significant at the bivariate level. We ran some additional regression analyses on other variables, such as prior incarceration, lifetime attempts to leave prostitution, anxiety, etc. but they were not significant. However, being a victim of childhood sexual abuse was significant in the adjusted analyses, so we changed the results throughout to reflect this. This was an interesting finding because the categories of CSE began at age 15 so the likelihood of childhood sexual abuse not overlapping with CSE is possible.

Reviewer Comment: Could the authors clarify whether they adjusted the multivariate regression on duration of trade sex? It is said in the Methods section but does not appear in any of the result tables. It could help to know what the median duration in sex trade is, whether it is very different between the two categories of women, and whether the association with experience of STI holds after taking the duration of sex trade into account.

Author response: Thank you for this important point. We re-ran the multiple logistic regression to adjust for duration of trading sex for the final model. Ever acquiring an STI became non-significant. We changed the tables and results/discussion according to the new results.

For median duration in sex trade: 3-5 years was the median (IQR: 1-3 years, 5-10 year) for everyone. [Data were collected in ranges, not years]. For CSE, it was the same). For non-CSE, it was less; 1-3 years was the median (IQR: 1-3 years, 3-5 years).

For ages 18-19, the median was lower: 6-12 months (IQR: 6-12 months, 1-3 years). For ages 19+, the median stayed the same as the overall median.

Reviewer Comment: The result which seems the most robust is the higher probability to have experienced CSE when one is having 5 or more clients a day in average. This is a well-known risk for STI and maybe the authors could insist more on this.

Author response: Thank you. We made sure to discuss this in our Discussion section, in the paragraph starting with Line 810.

For the Discussion:

Reviewer Comment: I think the Discussion really needs to be edited to avoid any misinterpretation of the results: In particular, line 225-226 it is said: “These individuals more commonly reported not operating independently, and those organizing their sex trade were police, similar to other studies intersecting with police in Russia”..why on the one hand the authors say that police were organizing the sex trade of women who entered as minors, and on the other hand, the same women declared less extortion from the police.

Author response: Thank you! We agreed the police interaction interpretations needed clarification. We omitted the line 225-226 “These individuals more commonly reported not operating independently, and those organizing their sex trade were police…” because it is not supported by the data. Quite the opposite, the women received greater assistance from police. We greatly rewrote the discussion section instead to make the police interactions clearer. [Lines 1186-1194].  We rewrote and re-organized the Discussion section to make all interpretations clearer, including the main findings.

Reviewer Comment: Another example line 238-239: “In this paper, those entering sex trade as minors were twice as likely to have an STI….” Those entering sex trade as minors were actually twice as likely to ever have an STI, which is not the same at all (please see above my comment on taking the duration in sex trade into account). 

Author response: The result did change when we adjusted for duration in the sex trade. STI was no longer significant, so we took that line out.

Reviewer Comment: Why would the women who experienced CSE have more clients a day than the others? Does that relate to different pathways into the sex trade?

Author response: Yes, we added to the discussion [starting on Line 810]: “Women who experienced CSE, compared to those who did not, may have taken a different pathway to prostitution after dropping out of the educational system early. Selling sex may be their only means of survival and income, resulting in their need for a higher number of clients per day in the sex trade. Although having more clients might result in a higher risk of contracting STIs, those who entered the sex trade as minors did not have significantly more STIs when adjusting for the length of time selling sex. However, research shows that Russian children who are sexually exploited are particularly vulnerable to STIs; By age 14, sexually exploited children in the Baltic study had a significantly higher incidence of syphilis and other STIs than adults [47]. Prohibition from using condoms or contraceptives might also better explain STI risk.”

Reviewer Comment: Could the authors clarify whether “drug use before 18” comprises alcohol?

Author response: We clarified in the Tables that “drug use before 18” did not include alcohol.

Reviewer Comment: There is a mistake on reference 44 line 264: the ref 44 does not correspond to a study in Russia.

Author response: The reference section has been updated and revised according to all of the changes and additions made to the revised manuscript. We checked all citations within the manuscript to make sure they lined up with the correct ones at the end.

Thank you again for all of your valuable feedback!

Sincerest regards,

Lianne A. Urada, Ph.D., M.S.W.

Assistant Professor, San Diego State University School of Social Work, Hepner Hall #119, 5500 Campanile Drive, San Diego, CA 92182-4119, Tel: 619-594-1712 Fax: 619-594-5991 lurada@sdsu.edu

&  University of California, San Diego, Division of Infectious Diseases and Global Public Health, Department of Medicine, and Center on Gender Equity and Health, La Jolla, CA, U.S.A. lurada@ucsd.edu

Maia Rusakova

St Petersburg University, St. Petersburg, Russian Federation, 199034, Universitetskaya Emb.,7-9; rusakova.maia@yandex.ru

Veronika Odinokova

St Petersburg University, St. Petersburg, Russian Federation, 199034, Universitetskaya Emb.,7-923

Round 2

Reviewer 1 Report

I have read through the paper and the reviewer comments and responses. I feel that the comments were very well addressed. I recommend accepting the paper now.

Reviewer 4 Report

I recommend accepting the paper now.